# [RE] It Is Not the Journey but the Destination: Endpoint Conditioned Trajectory Prediction

## 1 Reproducibility Summary

*The following paper is a reproducibility report for It Is Not the Journey but the Destination: Endpoint Conditioned Trajectory Prediction [5]. The basic code was made available by the author at this https url. To reproduce the rest of the ablation studies mentioned in the paper, we had to modify the model structure accordingly. The well-commented version of the code containing all ablation studies performed derived from the original code is available at this https url with proper instructions to execute experiments in ReadMe.*

**Scope of Reproducibility**

We have verified all claims made by the paper and results from different experiments mentioned in the paper to support the claims. The central claim of PECNet was to improve state-of-the-art performance on the Stanford Drone trajectory prediction benchmark by 20.9% and on the ETH/UCY benchmark by 40.8%.

**Methodology**

The PECNet model was trained on the drone dataset with social pooling at different conditioned points and on the ETH/UCY datasets without social pooling. Furthermore, the trained model was evaluated on the drone dataset at different values of evaluated samples. For the latter, GitHub was used as a reference with author-given code.

**Results**

Overall, we were able to reproduce all the results mentioned in the paper within 5% error compared to what was mentioned in the paper. 5% error is quite acceptable for this application, and this variation could be caused by setting the initial random seed before training

**What was easy**

Verification of the claims against the ETH/UCY benchmarks and Stanford drone benchmark trajectory prediction with the PECNet models was an easy task.

**What was difficult**

For the datasets of ZARA1 and ZARA2, there were gaps in the sequence of frames, and thus interpolation was done to ensure the continuity of way-points. This caused the ADE and FDE errors to increase. Also, to maintain common frequency for all the datasets, they were down-sampled accordingly. For the conditioned way-point positioning experiment (with and without ORACLE), ADE had to be calculated from 11 predicted positions to not alter the structure of the model, and FDE was also calculated from the 11th point. However, due to it, some ADE fluctuations after the sixth way-point (and later) were larger than the claimed results. Similar fluctuations were observed for FDE as well, but the relative trends support the paper's claim.

**Communication with original authors**

We have not contacted any of the original authors as all the results were reproduced satisfactorily.

# 1   Introduction

The paper reproduced in this report aims to tackle multiple pedestrian trajectory predictions using rich multi-modal predictions for the use of autonomous vehicles, social robots, etc. Earlier approaches to this problem have been auto-regressive in nature, i.e., using n points (or analogically, data from the last $t$ seconds) from the dataset to produce the immediately next point, and then this process recurs.

In this paper, the endpoint distribution conditioned on the past trajectory and the past trajectory features are modeled separately for each pedestrian. The future trajectory points are predicted based on the past and features from other pedestrians via social pooling. An assumption in this model is the absence of passive pedestrians or the fact that each pedestrian has an actual preconceived endpoint or destination and is motivated to reach there.

To formulate this report, we have experimented on the author's code by adding/removing social pooling layers, using truncation tricks, visualization tools, and changing between CVAE and VAE architectures to verify all the claims made by the author described in detail below. We also performed some experiments such as shifting origin to the current point, using different architecture for encoder and decoder networks with the hope of improving the results, which are also described in detail at the end.

# 2   Scope of reproducibility

The paper revolves around the claim that an important component of predicting the trajectory is the destination in multi trajectory forecasting. If the destination for the pedestrian is clear, then the trajectory can be easily resolved using a separate network that takes the past trajectory and the destination as input taking into account social interactions among fellow pedestrians. Hence the central idea and claim of the paper is to use Conditional Variational Auto Encoder (CVAE) to get the latent variable encoding conditioned on the destination from the ground truth, thus using the latent variable to infer the predicted destination, and also using it for predicting the rest of the future trajectory. We take $k$ samples of the latent variable for testing purposes to predict $k$ different admissible trajectories as output for different destinations derived from the latent encoding. The overall reduction in the value of best ADE (Average Displacement Errors) and FDE (Final Displacement Error) values for the Stanford Drone, ETH/UCY datasets by using the CVAE network is the central claim of the paper.

To support the argument that indeed given the destination, the rest of the predicted trajectory contributes much less error than the previous state of the art methods such as SGAN [3], which directly predict the future trajectory, the paper performs an ablation study where they give the ground truth of a way-point which they call as oracle instead of the best one from taking $k$ samples of the latent variable to get the decoupled error of predicting the trajectory. The results strongly support the argument.

Further, they also experimented with different values of $k$ to show that FDE tends to 0 as $k$ increases and ADE tends to a certain value, which also shows the decoupled error in predicting the rest of the trajectory.

This paper also introduces a non-local social pooling layer and a "truncation-trick," which improves diversity and multi-modal trajectory prediction performance.

Hence the claims can be summarized as follows:-

1. Conditioning the destination on the past trajectory using CVAE helps in explicit decoupling of the destination prediction and path prediction errors. It hence helps reduce the destination prediction error and the subsequent path prediction error.
2. Using the social pooling layer helps reduce the error in predicting the path given the history and the destination.
3. Using truncation trick, i.e., truncating the distribution for fewer values of $k$ from which samples are taken helps reduce the destination prediction error. Also, taking a higher sigma value for larger values of $k$ reduces the error.

# 3   Methodology

We used the GitHub repository provided by the author as the base. However, it only contained the base model for results on the drone dataset. In order to reproduce the rest of the experiments, we had to make changes accordingly.

## 3.1   Model descriptions

The base model used in the paper consists of 2 parts:

Starting with the past trajectory, the CVAE or Conditional Variational Auto Encoder part is used to get the representation of the latent variable conditioned on destination. The past trajectory after flattening is passed through an $E_{past}$ layer to get the past encoding. During training, the ground truth final destination is passed through the $E_{end}$ layer to get the destination encoding. The past and the destination encoding are concatenated and passed through the $E_{latent}$ layer to get the latent encoding distribution with dimension $\mathbb{R}^{n \times 2z_{dim}}$ (Where n is the no of vehicles in the batch and $z_{dim}$ is the hyperparameter denoting the size of the latent encoding) which characterize mean and variance of the latent encoding. At this stage, a latent encoding is sampled from this distribution and passed through the $D_{latent}$ layer to get the destination.

Second, the predictor network consists of social pooling layers and an MLP network to get the future trajectory. The predicted destination is concatenated with the past encoding and the absolute current position of the pedestrian with respect to a common global reference frame for all pedestrians. This concatenated encoding is passed through a series of social pooling layers which contain g, $\psi$ and $\theta$ networks masked by the social mask at each step to get the final future encoding. This future encoding is passed through the $P_{future}$ network to get the future trajectory with $t_f$ time steps.

The social mask is represented as a binary matrix $M \in \mathbb{R}^{n \times n}$ where n is the no. of vehicles in the batch. The value (i,j) is 1 in the matrix M if the $i^{th}$ and the $j^{th}$ vehicle come close to each other with a threshold distance $d$ in at least one of the time frames from their past trajectories for the frames they are observed. Refer to (3) where $F(.)$ denotes the frame number for that position.

The loss function used to train the model is given in (4). It consists of 3 terms. This first term is the KL divergence term to bring the distribution of the latent variable close to the required one, which is $N(0, 1)$. The second term is the reconstruction loss from CVAE, called the Average Endpoint Loss (AEL), and the last term is the Average Trajectory Loss (ATL), calculated as the sum of L2 losses between each of the predicted and ground truth future trajectory point.

The metric used for validation and testing is ADE and FDE. ADE is the Average Displacement Error and is calculated as the average of euclidean distances at all future time steps between predicted and ground-truth positions. While FDE is the Final Displacement Error (FDE), and it is the euclidean distance between the final predicted and ground truth positions of the future trajectory. Refer to eqn 1 and 2 for mathematical formulation of ADE and FDE. Here $\hat{u}_t$ refers to predicted trajectory position at time t, and $u_t$ is the ground truth trajectory position at time t.

A representative diagram of the network is given in Figure 1 and the architecture parameters for all the networks are shown in Table 1.

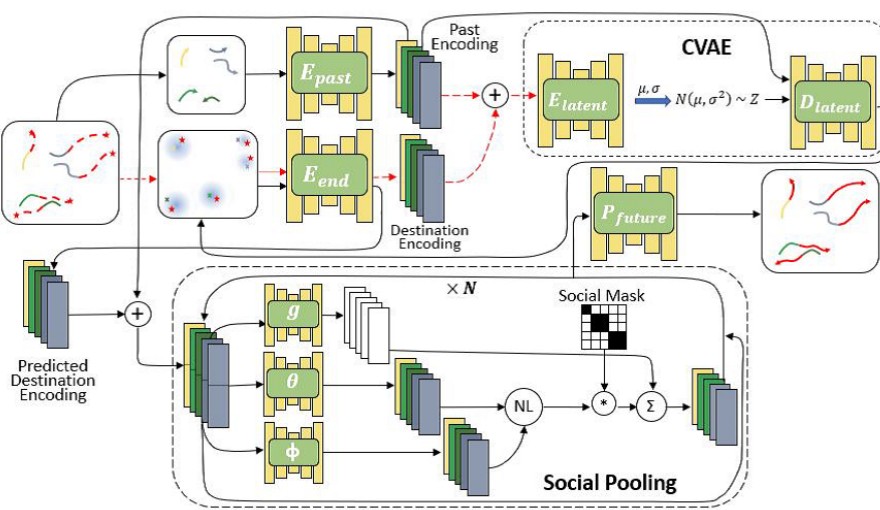

Figure 1: Model architecture [5]

$$ADE = \frac{\sum_{j=t_i+1}^{t_p+t_f+1} \|\hat{\mathbf{u}}_j - \mathbf{u}_j\|_2}{t_f} \tag{1}$$

$$FDE = \left\|\hat{\mathbf{u}}_{t_p+t_f+1} - \mathbf{u}_{t_p+t_f+1}\right\|_2 \tag{2}$$

| | Network Architecture |
|---|---|
| $\mathrm{E}_{way}$ | 2 -> 8 -> 16 -> 16 |
| $\mathrm{E}_{past}$ | 16 -> 512 -> 256 -> 16 |
| $\mathrm{E}_{latent}$ | 32 -> 8 -> 50 -> 32 |
| $\mathrm{D}_{latent}$ | 32 -> 1024 -> 512 -> 1024 -> 2 |
| $\theta, \Phi$ | 32 -> 512 -> 64 -> 128 |
| g | 32 -> 512 -> 64 -> 32 |
| $\mathrm{P}_{predict}$ | 32 -> 1024 -> 512 -> 256 -> 22 |

Table 1: Model Architecture [5]

$$\mathbf{M}[i,j] = \begin{cases} 0 & \text{if } \min_{1 \le m,n \le t_p} \left\| \mathbf{u}_m^i - \mathbf{u}_n^j \right\|_2 > t_{\text{dist}} \\ 0 & \text{if } \min_{1 \le m \le t_p} \left| \mathcal{F}\left(\mathbf{u}_0^i\right) - \mathcal{F}\left(\mathbf{u}_m^j\right) \right| * \min_{1 \le m \le t_p} \left| \mathcal{F}\left(\mathbf{u}_m^i\right) - \mathcal{F}\left(\mathbf{u}_0^j\right) \right| \right) > 0 \\ 1 & \text{otherwise} \end{cases} \tag{3}$$

$$\mathcal{L} = \lambda_1 \underbrace{D_{KL}(\mathcal{N}(\boldsymbol{\mu}, \boldsymbol{\sigma}) \| \mathcal{X}(0, \mathbf{I}))}_{KL \text{ Div in latent space}} + \lambda_2 \underbrace{\left\| \hat{\mathcal{G}}_c - \mathcal{G}_c \right\|_2^2}_{AEL} + \underbrace{\left\| \hat{\mathcal{T}}_f - \mathcal{T}_f \right\|^2}_{ATL} \tag{4}$$

## 3.2 Datasets

We used Stanford Drone [7] and ETH [6] / UCY [4] datasets. The Stanford drone dataset was given in the author's code, but ETH/UCY was not available. We took the processed ETH/UCY dataset from this https url which is available for open source use.

## 3.3 Hyperparameters

We used Hyperparameters given in the paper. We occasionally changed them accordingly, as mentioned in the paper, to perform the ablation studies described below. Mainly, the hyperparameters are $\sigma$: The variance used for sampling latent variable with mean 0, K: The no of guesses of final destination to make, $z_{dim}$: The size of latent encoding, and the model hyperparameters as explained above in model descriptions are summarized in Table 1.

## 3.4 Experimental setup

We used PyTorch to fluently conduct the aforementioned experiments. We extensively used Weights & Biases[1] for logging the experiments. For proper reproducibility, even after changing the machines, we set 42 as a system-wide seed before running every experiment. This helps in reproducing the exact results that are mentioned in the report.

## 3.5 Computational requirements

The proposed model can be trained on a single NVIDIA-K80 with 12 GB memory in less than an hour for both the Stanford Drone and ETH/UCY datasets. We were able to execute the experiments smoothly on Google Colab. Specifications of the machine are as follows:

NVIDIA-K80 GPU, Memory : 12 GB, Memory Clock : 0.82 GHz, Driver Version: 418.67, CUDA Version: 10.1

# 4 Results

The following experiments/ablation studies support the claims made earlier. The results are within 5% error from the ones claimed in the paper. We believe this much tolerance is acceptable in the context of this problem as results change by this variance on changing the random initial seed. A detailed description of the experiments and their results to support the claim are listed below:-

## 4.1 Experiment on the Stanford Drone Dataset (with and without social pooling, truncation trick)

The original model with hyperparameters, as mentioned in the paper, was trained and tested on the Standard Drone Dataset (SDD) [7]. The train-val-test set split is the standard split as described in [2], which is to preserve some scenes for test and validation and use others for training. We did it with social pooling and got results within 95% accuracy from claim results. The preprocessed data set for train and test were given on GitHub (by author). We used them to verify the results. Also, the truncation trick here refers to that $\sigma$ hyperparameter (Refer to hyperparameters section) is used as $c\sqrt{K-1}$ for K > 2 where c is a constant. The resultant distribution is truncated with $|z| < 1$ for sampling, meaning the sampling is done from a conditional Normal distribution conditioned on $|z| < 1$. Hence, the resulting sampled z from this distribution will always have $|z| < 1$. We did two experiments with hyperparameters, changing n-samples to 5 and another with n-samples to 20 as required for reproducing the results in the first table of the paper.

|  | O-S-TT | O-TT | Ours | PECNet-Ours |
|---|---|---|---|---|
| K | 20 | 20 | 5 | 20 |
| ADE | 10.56 / 10.47 | 10.23 / 10.19 | 12.79 /14.16 | 9.96/10.04 |
| FDE | 16.72 / 16.43 | 16.29 / 15.9 | 25.88 / 26.73 | 15.96/16.20 |

Table 2: Comparisons of our results against those of the authors' and previous state-of-the-art methods. -S' '-TT' represents ablations of our method without social pooling  truncation trick. We report results for in pixels for both K = 5  20 and for several other values of K. The format for each cell is <claimed result> / <reproduced result>

## 4.2 Experiment on ETH/UCY Datasets

ETH/UCY dataset consists of 5 scenes ETH, Hotel, Univ, Zara1, Zara2 extracted coordinates. We followed the conventional leave-one-out approach, i.e., trained on 4 sets and tested on the last set to get the results as was mentioned in the original paper. We verified results within 98% accuracy from claimed results. The occasional differences are understandable as the author did not mention the initial random seed, due to which there are small variations from claimed results which are understandable as they are within a 2% bound from the claimed results. The dataset was further down-sampled by 6 to get a 0.4 second gap between consecutive frames as demanded by the paper. The result is shown below in Table 3. With these two experiments, the reduction in error with respect to the previous results by using CVAE and subsequent reduction by using social pooling layer and truncation trick can be demonstrated.

|  | O-S-TT | | PECNet | |
|---|---|---|---|---|
| Datasets | ADE | FDE | ADE | FDE |
| ETH | 0.58/.57 | 0.96/.98 | 0.54/.53 | 0.87/.87 |
| HOTEL | 0.19/.20 | 0.34/.35 | 0.18/0.18 | 0.24/0.23 |
| UNIV | 0.39/0.32 | 0.67/0.53 | 0.35/0.32 | 0.60/0.49 |
| ZARA1 | 0.23/0.23 | 0.39/0.37 | 0.22/0.23 | 0.39/0.35 |
| ZARA2 | 0.24/0.20 | 0.35/0.33 | 0.17/0.20 | 0.30/0.32 |

Table 3: Quantitative results obtained versus those of the authors' (in the form of ours/authors'). 'O-S-TT' represents ablation of PECNet method without social pooling  truncation trick. The format for each cell is <claimed result> / <reproduced result>

## 4.3 Change in the structure of CVAE

In this experiment during training, the ground truth destination $(G_k)$ was used to predict the future $T_f$ instead of the one obtained from the latent variable during training. Hence the changes inside the code were to pass the ground truth $G_k$ and pass it to the social pooling layer during training. Hence, in this experiment, the training of the CVAE and the predictor networks are done separately decoupled from each other. Results of this experiment, as shown in Table 4, demonstrate that training on the latent encoding helps in coupling both parts of the network and improves the results. This newly trained network was tested on the Stanford drone dataset with social pooling, and we got results within 95% accuracy from the claimed results; again, the variation though very small, is due to the initial random seed difference.

|      | Claimed Result | Reproduced Result |
|------|----------------|-------------------|
| ADE  | 10.87          | 10.945            |
| FDE  | 17.03          | 16.277            |

Table 4: Change in the structure of CVAE

## 4.4 Effect of Number of samples (K)

We did this experiment on the Stanford drone dataset with social pooling. We trained the PECNet model with default $\sigma$ values of the CVAE and test on different k-sample values with and without truncation. Specifically, experiments were performed with changes in the hyperparameter $\sigma$ without truncation for k-sample <= 3, we used $\sigma$ with variance 1 and for k-sample > 3 we used $\sigma$ with variance 1.3. When using truncation trick for k-sample > 3, we used $\sigma$ with variance 1 and for k-sample <= 3 we used $\sigma$ with variance c * $\sqrt{k-1}$ as mentioned in the paper. In this experiment, we got results as shown in Table 5, within 95% accuracy from the claimed results with differences albeit small due to the differed random seed as it was not mentioned in the paper and with the same trend.

|              | 1     | 2      | 3     | 5      | 10    | 20    | 25    | 50    | 100   | 1000 | 10000 |
|--------------|-------|--------|-------|--------|-------|-------|-------|-------|-------|------|-------|
| ADE          | 24.29 | 18.457 | 16.25 | 14.16  | 12.04 | 10.49 | 10.06 | 8.99  | 8.208 | 6.81 | 6.27  |
| FDE          | 51.84 | 37.65  | 32.15 | 26.73  | 21.10 | 16.72 | 15.49 | 12.27 | 9.73  | 4.66 | 2.46  |
| Truncated-ADE| 17.62 | 16.67  | 15.71 | 14.788 | 12.10 | 10.21 | 9.74  | 8.54  | 7.70  | 6.39 | 6.02  |
| Truncated-FDE| 35.02 | 32.67  | 30.34 | 28.57  | 21.49 | 16.27 | 14.88 | 11.27 | 8.54  | 3.54 | 1.66  |

Table 5: Effect of no of samples (K) on ADE, FDE, Truncated-ADE, Truncated-FDE

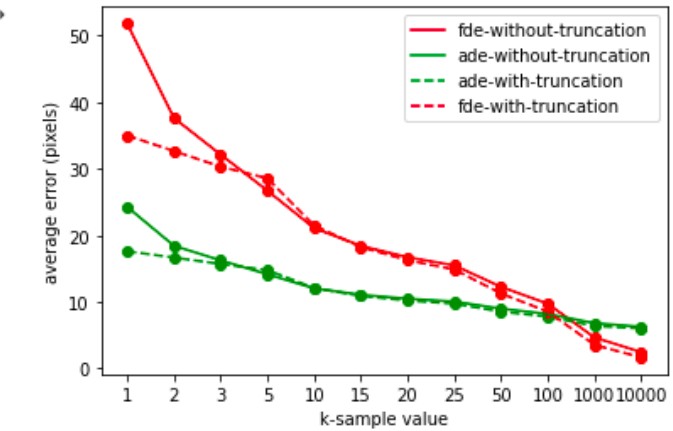

Figure 2: Graph of errors

## 4.5 Conditioned Way-point positions & Oracles

In this experiment, we conditioned on future trajectory points other than the last observed point, which we refer to as way-points. This was not clear in the paper about how to calculate FDE error because we can not predict the destination point according to the model architecture. We calculated FDE from the L2 difference between the last point of the predicted trajectory, as the final destination prediction is not available for this experiment. The observed result trends match exactly with the proposed results in the paper. It was done in two parts exactly as mentioned in the paper:

1. **With oracle**: During prediction of the future trajectory (at time of testing and validation), we gave ground-truth value of conditioned point instead of the best guessed one from sampling to predict trajectory from the model. The Stanford drone data set with social pooling and truncation trick was used to match with the results on paper. With this experiment, it can be demonstrated that the errors in destination prediction and path prediction

given the destination can be decoupled from each other. Hence using CVAE for inference on the destination as the first part helps in improving the results.

2. **Without oracle**: The same thing was done here, except during prediction of the future trajectory, the best guess for the conditioned point (predicted by the model) was taken (at time of testing and validation). Way-point Prediction Error was calculated as the difference between the ground truth of the conditioned point and the one predicted by the model. With this experiment, it can be empirically established that we get less inference error by conditioning on the destination point rather than on any of the intermediate points on its future trajectory.

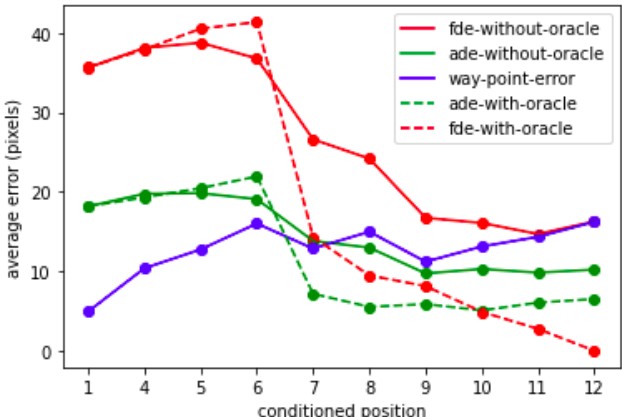

Figure 3: Graph of errors

|  | 1 | 4 | 5 | 6 | 7 | 8 | 9 | 10 | 11 | 12 |
|---|---|---|---|---|---|---|---|---|---|---|
| ADE | 18.16 | 19.76 | 19.83 | 19.08 | 13.82 | 12.98 | 9.73 | 10.29 | 9.83 | 10.218 |
| FDE | 35.64 | 38.125 | 38.77 | 36.79 | 26.61 | 24.18 | 16.73 | 16.08 | 14.69 | 16.27 |
| Way-point error | 4.93 | 10.38 | 12.75 | 16.01 | 12.86 | 14.98 | 11.207 | 13.12 | 14.336 | 16.23 |
| Oracle ADE | 18.17 | 19.30 | 20.46 | 21.94 | 7.17 | 5.52 | 5.87 | 5.074 | 6.0552 | 6.51 |
| Oracle FDE | 35.68 | 37.93 | 40.54 | 41.38 | 14.30 | 9.48 | 8.13 | 4.892 | 2.745 | 0.0 |

Table 6: Conditioned Way-point positions and Oracles

## 4.6 Reference shift (Extra experiment)

This is an extra experiment that we performed. The motivation behind this experiment is that the past trajectories passed as input are reference shifted with respect to the starting point of the past trajectory. We believe this would make it difficult for the social pooling layer to consider the social interaction impact on the future trajectory as their current position will be based on different reference frames for all the neighboring pedestrians on which the interaction would mostly depend. Instead, we experimented with setting the reference frame for past trajectory as the current position instead. Hence, passing the global current position and the relative past trajectory with respect to that frame for all pedestrians will be easier to learn from for the social pooling layer as social forces as it will be easy to infer global positions of all pedestrians at past frames from this setup.

We took the reference of the trajectory for each pedestrian as the current point instead of the first point of the past trajectory. This helped the CVAE network to get a better representation of the destination point as all past input trajectories have a common last point, which makes it easier for the encoder-decoder network to function; also, the predictor and social pooling network gets more easily trained. This experiment showed about 10% further decrease in ADE and FDE metrics for drone dataset as shown in the Table 7.

## 4.7 Using encoder and decoder LSTM network (Extra experiment)

The motivation behind this experiment is that using MLP for encoding the past trajectory and predicting the future trajectory won't implicitly leverage the advantage of the sequential nature of the past trajectory and future trajectory. Training using MLP is suitable for inputs with less and fixed history time frames, but would be difficult to tune for

| | Before Reference Shift | After Reference Shift |
|---|---|---|
| ADE | 9.96 | 8.64 |
| FDE | 15.96 | 14.63 |

Table 7: Results comparing before and after reference shift experiment for PECNet Model

larger history sizes. Hence, we experimented by using an LSTM network instead. This would also help to consider variable lengths of the past and future trajectory based on the requirement.

We used encoder LSTM instead of MLP to form the encoding of the past trajectory to accommodate the variable length of past trajectory and form a better representation as to the input temporal data. Also, we used the decoder LSTM network to predict the rest of the trajectory given the destination. However, the FDE error reduced by about 5%, but the ADE is surprisingly more, demonstrating that decoder LSTM does not perform well given the destination point.

| | Using MLP | Using LSTM |
|---|---|---|
| ADE | 9.96 | 26.9 |
| FDE | 15.96 | 14.3 |

Table 8: Results comparing using MLP v/s using LSTM for PECNet Model

## 5   Discussion

From each of the experiments, the claims made by the paper as described above can be strongly supported and empirically proved. The strong correspondence between destination and rest of the path is observed, as evident from the results in comparison to previous experiments. Also, the use of the social pooling layer and truncation trick reduces the error to a great extent, as demonstrated from the ablation studies described above. In order to further study the choice of structure of the network, two other experiments were performed described above, and they strongly support the choice of MLP architecture used for past encoding  future prediction instead of LSTM/GRU RNN structures.

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
