# OpenReview forum: "[RE] It Is Not the Journey but the Destination: Endpoint Conditioned Trajectory Prediction"
_ML_Reproducibility_Challenge/2020 — Reject_

### Official Review · AnonReviewer1 · 2021-03-01
**Reasonable efforts in reproducing the work based on open-sourced code**

**Rating:** 6
**Confidence:** 3

**Review:**

The report did a reasonable job in reproducing the research work on predicting future trajectories from past data by leveraging a goal-prediction generative model. The experiment demonstrated that the original work is quite reproducible and supports the claims made by the authors of the paper. It is also greatly appreciated that additional experiments are performed (reference shift and recurrent architecture) to further investigate the proposed idea. However,  I do have a few concerns regarding the report:

1. The majority of the experiments presented seem to be running with the existing code from the original authors. The report did mention that some changes are required for certain experiments but didn't provide more details. It would be helpful if the authors could provide more details regarding which experiments in the report required non-trivial changes to the codebase to better judge the effort required for the work.

2. Writing could be improved. For example, the definition of ADE and FDE should be made clear in the report (They are defined in Eq (1) and (2), which aren't referenced in the text.).


**Familiar With The Original Paper:**

I have not read the original paper

**Reproducibility Summary:**

Report has summary

---

### Official Review · AnonReviewer2 · 2021-03-02
**Review of "[RE] It Is Not the Journey but the Destination: Endpoint Conditioned Trajectory Prediction"**

**Rating:** 4
**Confidence:** 4

**Review:**

While there are some strengths, this report needs substantial improvement in a few areas.

Strengths
- The GitHub repository containing the code for reproducing the experiments appears to be thorough (although it breaks anonymity, and should have been anonymized before submission)
- There was an extra experiment on using an LSTM architecture instead of an MLP.
- The reproduction of the results from the original paper appears to be thorough.

Weaknesses
- If using the code from Mangalam et al., how do the authors account for the differences (albeit small) in the results?
- It is not clear how the GitHub repository for the paper reproduction differs from the original repository provided by Mangalam et al. without going through the commits. It appears that there are more comments and some additional experiments, but it would helpful if this were summarized in the report and the additional experiments were separated from the original repository in some way. This is made even more confusing by the fact that the README on the main page of the repository is unchanged from the original one (other than the "About" messsage on the page).
- The extra experiment should be clarified. Why did the authors chose to run it? What could it demonstrate?
- Figure 1 and Table 1 are directly from the Mangalam et al. paper, but this is not acknowledged.
- Notation (such as in Table 1) is not explained. Generally, this report is very difficult to read without cross-referencing the original paper.
- The same holds for key abbreviations used in the report but not explained, including FDE and ADE.

**Familiar With The Original Paper:**

I have read the original paper

**Reproducibility Summary:**

Report has summary

---

### Decision · Program_Chairs · 2021-03-31

**Decision:**

Reject

**Comment:**

Overall, the reproduction effort is fairly well done, but there aren't enough details about the data.